# Spinal needles versus conventional needles for fine-needle aspiration biopsy of thyroid nodules—A multicenter randomized controlled trial

Kasper Daugaard Larsen[1,2]*, Gitte Bjørn Hvilsom[2,3], Tobias Vennervald Andersen[2], Preben Homøe[2], Finn Noe Bennedbæk[4], Jens Pedersen[3,4], Lena Bjergved Sigurd[3,4], Jens Jessen Warm[1], Katalin Kiss[5], Giedrius Lelkaitis[5], Luise Andersen[5], Marie Røsland Rosenørn[6], Laszlo Hegedüs[7], Annette Kjær Ersbøll[8], Anne Fog Lomholt[1,3], Mikkel Kaltoft[1], Christoffer Holst Hahn[1,3], Tobias Todsen[1,3,9]

**1** Department of Otorhinolaryngology, Head and Neck Surgery and Audiology, Rigshospitalet – Copenhagen University Hospital, Copenhagen, Denmark, **2** Department of Otorhinolaryngology, Head and Neck Surgery, Zealand University Hospital, Køge, Denmark, **3** Institute of Clinical Medicine, Faculty of Health Sciences, Copenhagen University, Copenhagen, Denmark, **4** Department of Endocrinology and Metabolism, Herlev and Gentofte Hospital, Herlev, Denmark, **5** Department of Pathology, Rigshospitalet – Copenhagen University Hospital, Copenhagen, Denmark, **6** Department of Pathology, Zealand University Hospital, Roskilde, Denmark, **7** Department of Endocrinology, Odense University Hospital, University of Southern Denmark, Odense, Denmark, **8** National Institute of Public Health, University of Southern Denmark, Copenhagen, Denmark, **9** Copenhagen Academy for Medical Education and Simulation, Rigshospitalet, Copenhagen, Denmark

* kasper.daugaard.larsen@regionh.dk

## Abstract

### Objective

Ultrasound-guided fine-needle aspiration biopsy (FNAB) is essential for evaluating thyroid nodules but often yields inadequate samples, leading to repeated procedures, increased discomfort, and higher costs. Previous non-randomized studies found promising results of spinal needles to improve diagnostic adequacy. Therefore, we conducted a multicenter randomized controlled trial to validate these findings.

### Methods

Between July 1st, 2021, and April 13th, 2023, patients with suspicious thyroid nodules were randomized to receive FNAB with either a 25G spinal needle or conventional needle. The primary outcome was the rate of adequate diagnostic cytology. Secondary outcomes included procedure-related pain, sensitivity and specificity of FNAB, and adverse events.

**Data availability statement:** All de-identified individual-participant data that underpin the results in this article, the final trial protocol, and a full data dictionary/codebook are provided in the Supporting Information.

**Funding:** The study and first author (Kasper Daugaard Larsen) received funding from the Erik and Susanna Olesen's foundation and Copenhagen University Hospital – Rigshospitalet (no grant number). The funders had no role in study design, data collection and analysis, decision to publish, or preparation of the manuscript.

**Competing interests:** The authors have declared that no competing interests exist.

## Results

A total of 359 patients (75.6% female), with a mean age of 59.7 years (range 23−94) were randomized. The rate of adequate diagnostic FNAB was 86.2% (156/181) for the spinal group compared to 84.8% (151/178) for the control group (OR 1.01; 95% CI: 0.95–1.08). The mean pain scale score was 4.0 (SD = 1.8) in the spinal group and 3.9 (SD = 2.0) in the control group ($p = 0.40$). No complications were observed in either group. We found a significantly better cytological adequacy rate of FNABs performed by physicians with more than four years of experience in the procedure (OR=1.07; 95% CI, 1.01–1.14).

## Conclusions

No significant improvement was found using spinal needles with a stylet compared to conventional needles. Given the significantly higher cost of spinal needles and comparable diagnostic outcomes, their routine use for thyroid FNAB is not recommended.

## Introduction

Thyroid nodules are common in the general population, and prevalence increases with age, affecting up to 60% of elderly women [1,2]. While most thyroid nodules are benign, it is essential to rule out malignancy [3,4]. Ultrasound (US) of the thyroid gland allows a quantitative malignancy risk stratification of the thyroid nodule and the decision of whether a biopsy should be recommended [1,5,6]. However, variations in such FNAB recommendations to some extent depend on which US-based thyroid nodule risk classification system is used [7], as well as on lack of implementation of consensus in definition of US-based thyroid nodule descriptors [8,9]. US-guided fine-needle aspiration biopsy (FNAB) is recommended as a safe procedure and has high sensitivity and specificity [10,11]. However, 10–20% of FNABs from thyroid nodules yield inadequate samples [12]. In such cases, repeating FNAB is recommended, which increases patient discomfort, hospital costs, and delays the treatment of potential thyroid malignancies [13,14]. In case of repeat cytology remaining inadequate, a meta-analysis reported that diagnostic surgery occurred in 16.2% of cases [15].

To decrease the number of inadequate samples, some centers use a spinal needle with a stylet instead of a traditional needle for thyroid FNAB [16–18]. When using the spinal needle, the concept is to use the stylet to hinder contamination by tissue and blood until the needle reaches the nodule. A few studies have found that the spinal needles can significantly decrease the number of inadequate samples due to fewer blood contaminated FNAB smears [16,18]. However, these studies were non-randomized trials and relied on a single clinician performing all the FNABs [16–18].

The aim of the "SPInal needles versus conventional fine needles for ultrasound-guided fine Needle Aspiration Biopsy" (SPINAB) multicenter randomized trial [19] was to explore if these promising diagnostic results employing spinal needles for FNAB can be generalized to multiple centers, physicians with varying experience, and devoid of patient selection bias.

## Methods

Approvals have been granted by the Danish Research Ethical Committee (protocol #: 10613690) and the Danish Data Protection Agency (protocol #: P-076–2021). The protocol has been registered at the ClinicalTrials.gov database (NCT04879355) and subsequently published [19]. The ethical rules of the Helsinki Declarations, as revised in 2013, have been complied with and all patients provided verbal and written consent before inclusion. Our study follows CONSORT 2010 guidelines (S9 File). The individuals pictured in S2 Fig have provided written informed consent (as outlined in PLOS consent form) to publish their image alongside the manuscript.

### Study design and population

The study was an investigator-initiated, multicenter, single-blinded, randomized controlled trial. The study population consisted of patients ($\geq$18 years), who were referred for thyroid FNAB at one of three departments between July 1st, 2021, and April 13th, 2023, and were eligible for enrolment according to our inclusion and exclusion criteria (S1 File). The diagnostic work-up and patient inclusion were conducted at the outpatient clinics at three departments in Denmark:

- Department of Otorhinolaryngology, Head and Neck Surgery. Included patients for thyroid nodule assessment, patients in the Danish "Cancer Fast Track" and patients referred for therapeutic intervention, such as radiofrequency ablation.

- Department of Endocrinology. Included patients for thyroid nodule assessment.

- Department of Otorhinolaryngology, Head and Neck Surgery and Audiology. Included patients in the Danish "Cancer Fast Track" and patients referred for therapeutic intervention, such as radiofrequency ablation.

### Interventions

Patients who accepted participation were randomized (1:1 ratio) to undergo FNAB with either a 25G spinal needle (spinal group) or a 25G conventional fine needle (control group). If a patient had multiple thyroid nodules qualifying for biopsy, only the nodule with the highest EU-Thyroid Imaging Reporting and Data Systems (EU-TIRADS) score was biopsied as the randomized nodule in the present study [20].

The biopsy was performed with a 25G (0.515 mm) and 50 mm long spinal needle (BD Quincke Spinal Needle 25G x 1", Neonatal, Becton Dickinson, Franklin Lakes, New Jersey, USA) in the spinal group, and with a 25G and 40 mm long conventional needle (Braun Sterican Orange needle 25G x 1,5", Braun Melsungen AG, Melsungen, Germany) in the control group. To ensure correct handling of the needle and removal of the stylet, all physicians (minimum one year of experience with FNAB) received instructions followed by supervision in ultrasound-guided FNAB on a phantom. The physicians were further instructed to use the capillary technique in all cases [21]. In the spinal group the stylet was removed within the nodule before performing the biopsy (S2 File). Aside from that the same FNAB technique was used regardless of randomization.

### Outcomes

The primary outcome was the rate of adequate diagnostic cytology (Categories II-VI in the Bethesda System for Reporting Thyroid Cytopathology (BSRTC)) of FNABs from thyroid nodules [22]. Subgroup analysis was performed on needle type, nodule solidity, and the experience level of the physicians performing the FNABs (more than four years OR four and less years of experience with the US-guided FNAB procedure).

The secondary outcomes were the procedure-related pain, adverse events, sensitivity and specificity of the FNABs. The procedure-related pain was measured with the Numerical Rating Scale (NRS) from 0–10, quantified by the patient directly following the procedure [23]. Acute complications, such as bleeding, hematomas, (pre)syncope, and recurrent

laryngeal nerve injury, were recorded immediately after the procedure. The patients' medical records were reviewed for admissions or contacts concerning late complications such as infections, and hematomas requiring treatment, 6 months after the FNAB procedure. Nodule characteristics, such as nodule solidity (solid/cystic-solid) and largest nodule diameter (mm), were based on the US observations. Pathology reports from patients who underwent thyroid surgery during the trial were used as a diagnostic reference (i.e., gold standard) when calculating the sensitivity and specificity of the FNAB evaluations (S3 File).

## Cytopathological evaluation

All slides were air-dried, and May-Grünwald Giemsa was used for the staining at both Departments of Pathology. A group of five head and neck pathologists, blinded to the randomization, evaluated all the smears. Criteria for adequacy were based on the BSRTC. According to these criteria, a minimum of six well-visualized follicular groups, each with a minimum of 10 cells, on one glass was required for sample adequacy. Exceptions to this numerical requirement were samples containing abundant colloid, atypia, or whenever a specific diagnosis could be made [22]. Thus, samples classified as BSRTC Category II-VI were considered adequate, while BSRTC Category I samples were considered inadequate/non-diagnostic. In case of inadequacy (BSRTC Category I), the specimen was reexamined independently by another pathologist, and a final diagnosis was made based on reaching consensus.

The diagnostic correlation with histology was obtained by comparing FNAB results with histology in those patients who underwent subsequent surgery (S3 File).

## Randomization and allocation concealment

Participants were randomly assigned (1:1) to the spinal needle group or the conventional needle group using a computer-generated random sequence with permuted block sizes (=10). The random sequence was generated by two investigators before the start of the trial using the website www.randomization.com. To ensure strict allocation concealment, the group assignments were sealed in sequentially numbered, opaque envelopes prepared before the start of the trial. Each envelope contained the designated needle (spinal or conventional) and was only opened by the clinician after the patient had been enrolled.

## Blinding

This trial was single-blinded. The patients were not informed about needle type used for their biopsy, and importantly, the cytopathologists were blinded as for the needle type. The clinicians performing the FNAB procedure, however, could not be blinded due to the obvious differences in needle appearance and handling, these operators were aware of the needle type upon opening the envelope. No other personnel had access to allocation information during the trial.

## Sample size

The sample size was determined based on an expected 10% absolute increase in adequate cytological diagnoses (from 86% in the control group to 96% in the spinal needle group, as suggested by prior studies [16,18]. Using a two-sided α of 0.05 and power (1–β) of 90%, we calculated that 175 patients per group (350 total) would be required to detect this difference [19].

The calculated sample size (n = 350) referred to participants who actually underwent the allocated FNAB. To compensate for anticipated post- randomization drop-outs (~10%), we continued enrolment until ≥ 350 procedures had been completed. Enrollment stopped after 380 envelopes had been opened; 21 patients did not proceed to, leaving 359 analyzable participants.

## Statistical analysis

A statistical analysis plan was agreed upon before evaluating the data [19].

Continuous variables are presented as means and standard deviations, and categorical variables as frequencies and percentages.

A binary logistic regression was used to compare cytological results (adequate/inadequate) obtained using the two types of needles. To adjust for the multicenter design, analyses of the primary outcome were conducted using a generalized estimating equation (GEE) logistic regression model with study center as a clustering variable. Similarly, clustering by operator was considered to account for any physician-related effects. The results are presented as odds ratios (OR) with 95% confidence intervals (95% CI).

Differences in outcomes between the two types of needles were examined using a mixed model linear regression (nodule dimension and pain scale score), a binary logistic regression with a generalized estimating equation (nodule solidity, complications and use of anti-coagulant/anti-platelet drugs) and a multinomial logistic regression with a generalized estimating equation (EU-TIRADS and BSRTC categories). If the assumptions for performing a linear regression were not fulfilled, the analysis was repeated with a rank transformation of the outcome.

For the primary outcome, subgroup analyses were performed for type of needle, nodule solidity, and experience level of the physicians performing the FNABs.

Characteristics analyzed included sex, age, largest nodule diameter measured by US, plasma levels of thyroid stimulating hormone (TSH), nodule solidity, clinical findings, EU-TIRADS score, BSRTC category, anticoagulant use and previous health care history.

The overall trial alpha was fixed at 0.05 for the single, prespecified primary outcome. Secondary and subgroup analyses were exploratory; accordingly, their p-values are presented without multiplicity adjustment and any findings should be regarded as hypothesis-generating.

After trial completion, we compared patient characteristics (age, sex) between the study sample and the patients who underwent FNAB in routine care without being invited or declined participation in the study to test for selection bias and evaluate the generalizability of our data (S4 File).

All analyses used the dataset provided in S7 File and variable definitions are detailed in the codebook S8 File.

A p-value of < 0.05 was considered statistically significant. Data were analyzed using Statistical Analysis Software, SAS, version 9.4 (SAS Institute Inc., Cary, NC, USA).

## Results

Between July 1st, 2021, and April 13th, 2023, 1,479 patients were referred to the three departments for thyroid FNAB. Of these, 380 (25.7%) were randomized, however, for unknown reasons 21 patients did not receive the allocated intervention (Fig 1). A total of 359 patients received the allocated intervention, 169 patients (47.1%) at the Department of Otorhinolaryngology, Head and Neck Surgery, 116 patients (32.3%) at the Department of Endocrinology and 74 patients (20.6%) at the Department of Otorhinolaryngology, Head and Neck Surgery and Audiology 181 were randomized to the spinal group and 178 to the control group. The mean age of the cohort was 59.0 years (range 23–94), with 75.6% being female. The baseline clinical characteristics were comparable between the two groups (Table 1).

The majority (86%) of the patients had a cold nodule on scintigraphy. The mean largest diameter of the nodules was 31 mm (SD,14 mm) in the spinal group and 31 mm (SD,13 mm) in the control group. Comparing the two groups, no differences were observed regarding nodule solidity, EU-TIRADS classification or the use of anticoagulant/anti-platelet drugs (Table 1).

Among the 359 patients, the overall FNAB diagnostic adequacy rate was 85.5% (n = 307). The rate of cytological adequacy was 86.2% (156/181) in the spinal group compared to 84.8% (151/178) in the control group (OR 1.01, 95% CI: 0.95–1.08, p = 0.74) (Fig 2). The most common causes for inadequate cytology were insufficient number of follicular cells (44.2%), obscuring blood (23.1%), and cyst fluid only (17.3%) (Table 2). The reasons for inadequate samples were similar between the two groups, except for six samples (11.5%) with clotting artifacts seen in the spinal group compared to none (0%) in the control group (p = 0.009).

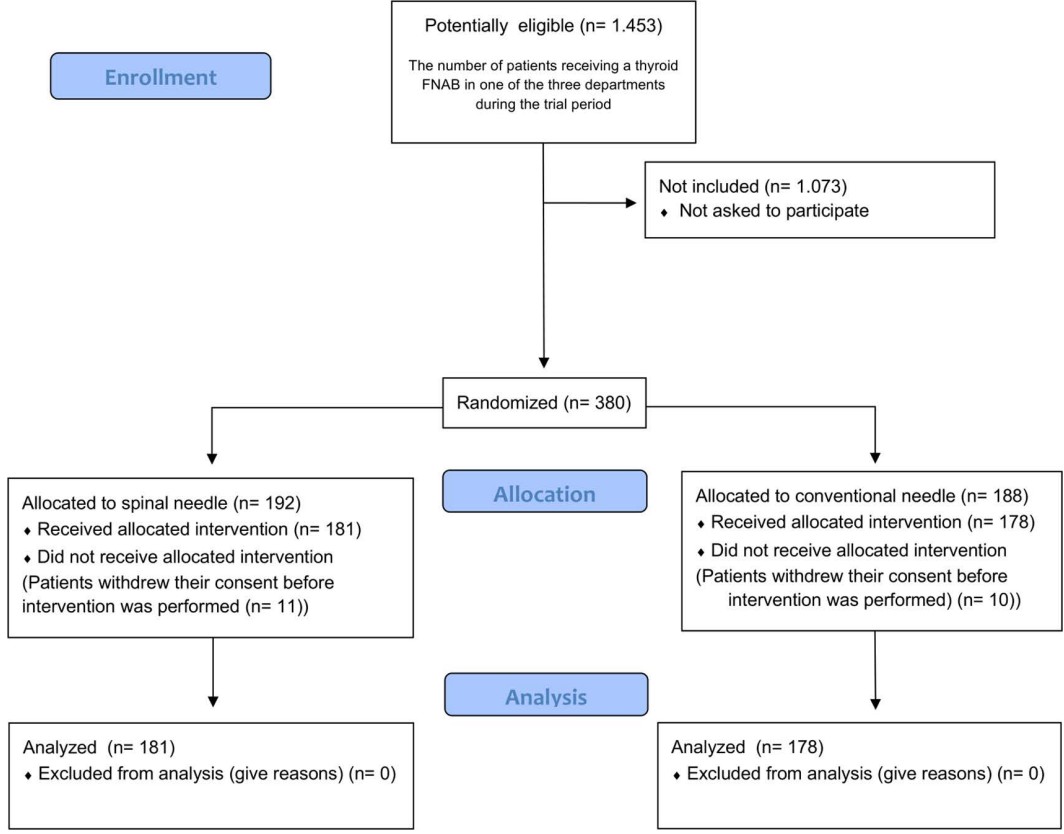

**Fig 1. CONSORT diagram illustrating the study flow.** *Abbreviation: FNAB, Fine-needle aspiration biopsy.*

A benign cytology (Bethesda category II) was observed in 70.2% (127/181) of the spinal group's FNABs versus 68.8% (122/178) in the control group. Malignant cytology (Bethesda category VI) was observed in 2.8% (10/359) of the samples, six (3.3%) in the spinal group and four (2.3%) in the control group. There were no differences in the distribution of samples in the different Bethesda categories between the two groups ($p = 0.90$) (S5 File).

No significant difference in mean NRS pain score was found between the spinal (4.0; SD, 1.8) and the control group (3.9; SD, 2.0) ($p = 0.40$; (Table 2)).

No complications, such as infections, bleeding, hematomas, (pre)syncope or recurrent laryngeal nerve injury, were observed in either group during the trial.

In the subset of participants (n = 76) who underwent surgery, 22 had histologically confirmed carcinomas and 54 had benign lesions. The FNAB sensitivity in the spinal group was 88.9% (95% CI: 56.5–98.0), compared to 85.7% (95% CI: 48.7–97.4) in the control group ($p = 1.0$). The FNAB specificity was 68.8% (95% CI: 44.4–85.8) in the spinal group compared to 80.0% (95% CI: 49.0–94.3) in the control group($p = 0.86$) (See S5 File for details).

In our subgroup analyses (Table 3), we found the odds of achieving adequate cytology to be statistically significantly higher for solid nodules than for cystic-solid nodules (OR=1.11; 95% CI: 1.05–1.17). Additionally, physicians with more than four years of experience in the procedure had a significantly higher proportion (88.9%) and odds of achieving adequate cytology compared to physicians with four years or less (78.3%) across all three centers (OR 2.12; 95% CI: 1.03–4.38). Although indicating a trend regarding influence of experience, the cytological adequacy for those with four years of experience or less was 79.6% in the spinal group and 77.1% in the control group

**Table 1. Characteristics of the study population.**

| Characteristics | | Randomization – type of needle | |
|---|---|---|---|
| | Total N = 359 | Spinal needle N = 181 | Conventional needle N = 178 |
| Female, n | 268 (74.5) | 134 (74.0) | 134 (75.3) |
| Age (Years), mean (SD) | 359 | 59.5 (14.2) | 59.9 (14.2) |
| Nodule dimension, mm, mean (SD) | | 31 (14) | 31 (13) |
| TSH, mE/l, mean (SD) | 321 | 1.91 (8.1) | 1.70 (2.4) |
| Nodule solidity | | | |
| Cystic-solid nodule (<50% cystic) | 167 (46.9) | 92 (51.5) | 75 (42.4) |
| Solid nodule | 189 (53.1) | 87 (48.6) | 102 (57.6) |
| Clinical findings | | | |
| Pressure symptoms | 156 (43.4) | 80 (44.2) | 76 (42.7) |
| Cold nodule on scintigraphy | 307 (85.6) | 151 (83.4) | 156 (87.6) |
| FDG-PET-positive thyroid nodule | 39 (10.9) | 23 (12.7) | 16 (9.0) |
| Suspicious high-risk clinical findings[a] | 22 (6.1) | 9 (5.0) | 13 (7.3) |
| Other[b] | 121 (33.4) | 65 (35.9) | 56 (31.5) |
| EU-TIRADS category | | | |
| EU-TIRADS 2 | 10 (2.8) | 7 (3.9) | 3 (1.7) |
| EU-TIRADS 3 | 216 (61.2) | 112 (62.9) | 104 (59.4) |
| EU-TIRADS 4 | 68 (19.3) | 33 (18.5) | 35 (20.0) |
| EU-TIRADS 5 | 59 (16.7) | 26 (14.6) | 33 (18.9) |
| Use of anti-coagulant/anti-platelet drugs | | 27 (15.0) | 31 (17.4) |

*Note: Values are numbers (%) if nothing else is indicated.*

[a] *Either suspected neck lymph node or rapid thyroid nodule growth, hard and immobile nodule, suspicion of recurrent laryngeal nerve palsy.*

[b] *Either cosmetic concerns, dysphagia, pain or CT-finding.*

*Abbreviation: TSH, Thyroid Stimulating Hormone; FDG-PET, Positron Emission Tomography; EU-TIRADS, European Thyroid Imaging and Reporting Data System.*

($p = 0.73$). For those with more than four years of experience the cytological adequacy was 89% in the spinal group and 88.9% in the control group ($p = 0.98$). There was no statistically significant difference in the adequacy rate in the spinal and control group FNABs for either experience level (see S6 File for the distribution of operator expertise).

## Discussion

In this multicenter randomized clinical trial we demonstrate that spinal needles do not improve the rate of cytologically adequate FNABs compared to conventional needles. Similar mean pain scale scores for both the spinal and conventional needle groups were observed, and no complications encountered during the trial period. Furthermore, no statistically significant difference in cytological adequacy was observed between the spinal and control group FNABs, regardless of the physicians' experience levels and nodule solidity.

Major strengths of this study are the randomized design and adequate sample size, based on a power analysis, including 359 patients from three different hospitals in the final analysis. The physicians performing the FNAB in the SPINAB trial had varying experience levels and came from both Endocrinology and ENT departments. Thus, we applied appropriate statistical methods to account for clustering at hospital and physician experience level in our analysis. Based on the composition of physicians and their level of experience being the same in the spinal and the control group, we argue that

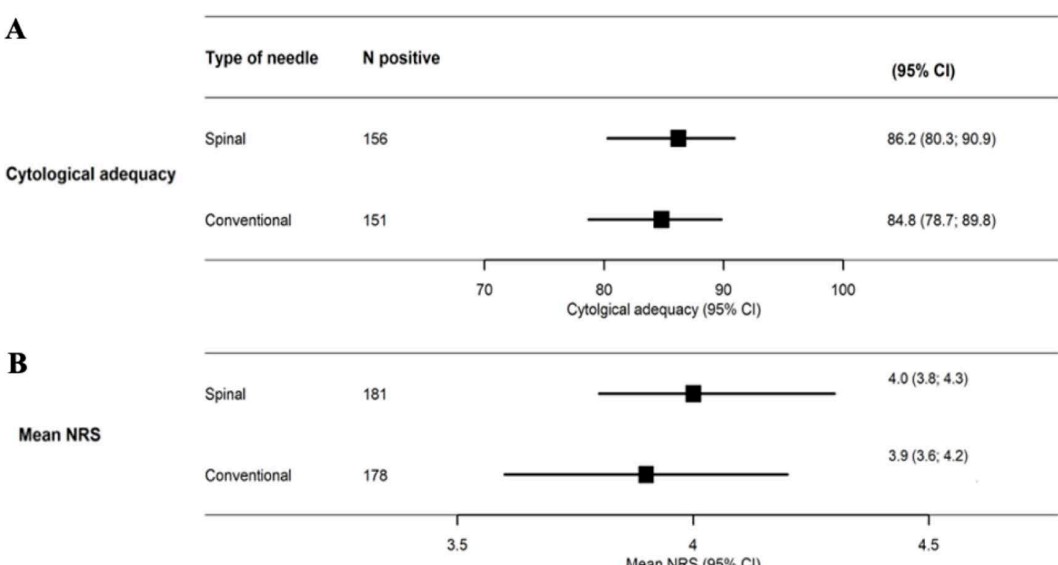

**Fig 2. (A) Cytological adequacy rate for the spinal needle and conventional needle FNABs. (B)** Mean NRS pain score for the spinal and conventional needle groups. *Abbreviation: FNAB, Fine-Needle Aspiration Biopsy; NRS, Numerical Rating Scale.*

**Table 2. Comparison of cytological findings, pain score and complications according to needle type.**

| | | Spinal needle | Conventional needle | p-value |
|---|---|---|---|---|
| Cytological adequacy, n (%) | | 156 (86.2) | 151 (84.8) | 0.74 |
| NRS pain score, mean (SD) | | 4.0 (1.8) | 3.9 (2.0) | 0.40 |
| Complication | | 0 (0.0) | 0 (0.0) | – |
| Bethesda I (n = 52) – Reason for inadequate sample | | | | |
| Cyst fluid only | 9 (17.3) | 3 (12.0) | 6 (22.2) | 0.47 |
| Acellular specimen | 2 (3.9) | 0 (0.0) | 2 (7.4) | 0.49 |
| Obscuring blood | 12 (23.1) | 5 (20.0) | 7 (25.9) | 0.75 |
| <60 follicular cells | 23 (44.2) | 11 (44.0) | 12 (44.4) | 1.0 |
| Artifact from clotting | 6 (11.5) | 6 (24.0) | 0 (0.0) | 0.009 |

*Abbreviation: NRS (Numerical Rating Scale).*

our results might be generalized to other clinical settings where FNAB is performed. Another strength is the use of two independent pathologists to evaluate all inadequate FNAB samples to ensure a higher validity of our reference test.

This study has several limitations that warrant consideration. First, pain was assessed with a single, patient-reported 0–10 Numerical Rating Scale immediately after biopsy. Such subjective measurement is susceptible to individual pain-threshold variation. Second, although we adjusted for operator experience (> 4 years vs ≤ 4 years) in the multivariable models, other aspects of operator variability (e.g., specialty background, annual biopsy volume, junior vs senior status) could not be fully captured and may still influence cytological adequacy. Third, blinding was incomplete, patients and cytopathologists were blinded to needle type, but the performing clinicians could not be blinded because of obvious visual and tactile differences between the needles, introducing a potential performance bias. Fourth, A limitation of our study is that only 26.1% of patients referred for thyroid FNAB at the participating hospitals were enrolled in the study. This is mainly due to the SPINAB trial being conducted in a real-world clinical setting, where limited time and availability did

**Table 3. Subgroup analyses of cytological adequacy.**

| Predictors | N | N_events (%) | OR (95% CI) |
|---|---|---|---|
| Type of needle§ | | | |
| Spinal needle | 181 | 156 (86.2) | 1.01 (0.95; 1.08)[1] |
| Conventional needle | 178 | 151 (84.8) | 1 (ref) |
| Nodule solidity£ | | | |
| Cystic-solid nodule | 167 | 134 (80.2) | 1 (ref) |
| Solid nodule | 189 | 170 (90.0) | 1.11 (1.05; 1.17)[2] |
| Operator experience level# | | | |
| ≤4 years of experience | 115 | 90(78.3) | 1 (ref) |
| > 4 years of experience | 244 | 217 (88.9) | 2.12 (1.03; 4.38)[3] |

§ *adjusted for hospital and physician within hospitals.*

£ *adjusted for the type of needle, hospitals, and physician within hospitals.*

# *adjusted for the type of needle and hospitals.*

not allow all patients to be invited. Further, only physicians who received instructions and hands-on training in the use of spinal needle FNAB could include patients. We prioritized the training of physicians in the spinal needle technique to ensure standardized performance in the spinal group. Consequently, we could not include all patients referred for FNAB. However, since our post hoc analysis did not find any differences between the included vs. non-included patients in the SPINAB trial, we believe the results are generalizable beyond the included patients (S4 File).

To our knowledge, this study is the first randomized clinical trial investigating the use of spinal needles for US-guided thyroid FNAB. Prior to the current study, three non-randomized prospective studies by Cappelli et al. and Ahari et al. have been performed, including 516, 320, and 240 patients, respectively [16–18]. These studies found that spinal needles reduce inadequate samples by 6–12 percentage points [16–18]. However, those studies were single-center, relied on a single operator and lacked randomization, raising the possibility of selection and performance biases. Our multicenter RCT, involving 23 operators across three hospitals, provides the first high-level evidence that spinal needles do not confer a clinically relevant advantage when rigorous randomization and blinding of outcome assessment are applied.

Cappelli et al. observed a reduction of inadequate samples due to blood contamination, from 18.7% using conventional needles to 3.1% using spinal needles with a stylet [18]. In our study, however, no difference in the rate of inadequate samples due to blood contamination was observed. A possible reason for the discrepancy in findings may be that participants in previous studies were not randomized to needle type, introducing a risk of selection bias. Patients with certain characteristics, such as increased vascularity at the sampling site, may have been more likely to receive one type of needle over the other, influencing the rate of blood contamination. Additionally, in our study the primary cause for an inadequate sample was a lack of sufficient number of follicular cells, with no significant difference between spinal and conventional needles. This indicates that spinal needles do not address the primary challenge of obtaining an adequate number of follicular cells, which aligns with expectations given that both needles have the same inner diameter.

There were no differences between spinal needles and conventional needles in identifying malignant and benign nodules, when compared in the 21.2% (n = 76) undergoing surgery. Our results indicate that FNAB overall demonstrates a sensitivity of 87.5% and specificity of 73.1%, which are within the ranges of previous studies [24,25]. However, it is important to clarify that the aim of this study was not evaluate the sensitivity and specificity of FNAB, as it was not powered for such an analysis. Furthermore, because only one primary comparison was powered and multiplicity adjustment was not applied to exploratory analyses, positive signals among secondary outcomes should be interpreted with caution.

Previous reports have demonstrated that an operator's experience is fundamental and depends both on the skills acquired using US and the ability to adequately prepare the biopsy specimen for cytopathology [26,27]. Our data confirm this, as we observed a significant association between years of experience with the procedure and diagnostic adequacy after controlling for factors such as needle type and hospital (OR 2.12; 95% CI: 1.03–4.38). The European Federation for Ultrasound in Medicine and Biology (EFSUMB) also recommends competency-based training to ensure a consistent basic level of skills among physicians performing FNAB [28].

Our findings indicate that spinal needles with a stylet do not significantly improve the diagnostic adequacy of cytological samples compared to conventional needles. Given that both needle types showed similar rates of cytological adequacy, patient-reported pain scores, and safety profiles, we conclude that the choice of needle, spinal or conventional, does not substantially impact the quality of FNAB outcomes.

Notably, we have shown that the conventional needle, which in most countries costs around 50 times less than the spinal needle, provides the same adequacy rate for FNAB cytology [18]. Therefore, given their higher cost and the lack of improvement in diagnostic outcomes, our results do not support change in current clinical practice. Instead, the focus should be on ensuring that clinicians performing FNAB receive adequate training and acquire sufficient experience to achieve optimal diagnostic outcomes. However, validated assessment tools to evaluate FNAB competence are lacking and ought to be implemented and employed in future studies.

## Supporting information

**S1 File. Eligibility criteria.**
(DOCX)

**S2 File. FNAB technique.**
(DOCX)

**S3 File. Sensitivity and specificity (S3a and S3b Tables).**
(DOCX)

**S4 File. Test for selection bias (S4 Table).**
(DOCX)

**S5 File. Bethesda categories in the spinal and control groups (S5 Table).**
(DOCX)

**S6 File. Expertise level at inclusion sites (S6 Table).**
(DOCX)

**S7 File. SPINAB_trial_dataset.**
(XLSX)

**S8 File. Codebook.**
(DOCX)

**S9 File. CONSORT 2010 checklist of information to include when reporting a randomised trial.**
(DOCX)

## Acknowledgments

The authors would like to thank Tina Klitmøller Agander for her contributions to the cytological evaluations.

## Author contributions

**Conceptualization:** Kasper Daugaard Larsen, Gitte Bjørn Hvilsom, Tobias Vennervald Andersen, Laszlo Hegedüs, Tobias Todsen.

**Data curation:** Kasper Daugaard Larsen, Gitte Bjørn Hvilsom, Tobias Vennervald Andersen, Preben Homøe, Jens Pedersen, Lena Bjergved Sigurd, Annette Kjær Ersbøll, Mikkel Kaltoft, Christoffer Holst Hahn, Tobias Todsen.

**Formal analysis:** Kasper Daugaard Larsen, Annette Kjær Ersbøll.

**Funding acquisition:** Kasper Daugaard Larsen, Tobias Vennervald Andersen, Tobias Todsen.

**Investigation:** Kasper Daugaard Larsen, Gitte Bjørn Hvilsom, Tobias Vennervald Andersen, Preben Homøe, Finn Noe Bennedbæk, Jens Pedersen, Lena Bjergved Sigurd, Jens Jessen Warm, Katalin Kiss, Giedrius Lelkaitis, Luise Andersen, Marie Røsland Rosenørn, Anne Fog Lomholt, Mikkel Kaltoft, Christoffer Holst Hahn.

**Methodology:** Kasper Daugaard Larsen, Gitte Bjørn Hvilsom, Tobias Vennervald Andersen, Preben Homøe, Finn Noe Bennedbæk, Katalin Kiss, Giedrius Lelkaitis, Luise Andersen, Marie Røsland Rosenørn, Laszlo Hegedüs, Annette Kjær Ersbøll, Tobias Todsen.

**Project administration:** Kasper Daugaard Larsen, Gitte Bjørn Hvilsom, Tobias Vennervald Andersen, Jens Pedersen, Jens Jessen Warm.

**Resources:** Kasper Daugaard Larsen.

**Software:** Kasper Daugaard Larsen, Tobias Vennervald Andersen, Annette Kjær Ersbøll, Tobias Todsen.

**Supervision:** Gitte Bjørn Hvilsom, Laszlo Hegedüs, Tobias Todsen.

**Validation:** Kasper Daugaard Larsen, Laszlo Hegedüs, Tobias Todsen.

**Visualization:** Kasper Daugaard Larsen.

**Writing – original draft:** Kasper Daugaard Larsen.

**Writing – review & editing:** Kasper Daugaard Larsen, Gitte Bjørn Hvilsom, Preben Homøe, Finn Noe Bennedbæk, Jens Pedersen, Lena Bjergved Sigurd, Katalin Kiss, Giedrius Lelkaitis, Luise Andersen, Marie Røsland Rosenørn, Laszlo Hegedüs, Annette Kjær Ersbøll, Anne Fog Lomholt, Mikkel Kaltoft, Christoffer Holst Hahn, Tobias Todsen.

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
