## [Decision Letter · Decision Letter 0]

Dear Dr. Larsen,

Thank you for submitting your manuscript to PLOS ONE. After careful consideration, we feel that it has merit but does not fully meet PLOS ONE’s publication criteria as it currently stands. Therefore, we invite you to submit a revised version of the manuscript that addresses the points raised during the review process.

We look forward to receiving your revised manuscript.

Kind regards,

Shafiya Imtiaz Rafiqi, PhD

Academic Editor

PLOS ONE

Journal Requirements:

[The study and first author (Kasper Daugaard Larsen) received funding from the Erik and Susanna Olesen’s foundation and Copenhagen University Hospital – Rigshospitalet (no grant number).]. 

3. Thank you for stating the following in your manuscript:

[The study and first author (Kasper Daugaard Larsen) received funding from the Erik and Susanna Olesen’s foundation and Copenhagen University Hospital – Rigshospitalet (no grant number).]

[The study and first author (Kasper Daugaard Larsen) received funding from the Erik and Susanna Olesen’s foundation and Copenhagen University Hospital – Rigshospitalet (no grant number).]

[No competing financial interests exist.].

5. In the online submission form, you indicated that [The data underlying this trial will be shared on reasonable request addressed the corresponding author.].

7. We note that Supporting Figure S2: FNAB Technique includes an image of a participant in the study.

Reviewers' comments:

Reviewer's Responses to Questions

**Comments to the Author**

1. Is the manuscript technically sound, and do the data support the conclusions?

Reviewer #1: No

Reviewer #2: Yes

Reviewer #3: Partly

2. Has the statistical analysis been performed appropriately and rigorously?

Reviewer #1: No

Reviewer #2: Yes

Reviewer #3: Yes

3. Have the authors made all data underlying the findings in their manuscript fully available?

Reviewer #1: No

Reviewer #2: Yes

Reviewer #3: Yes

4. Is the manuscript presented in an intelligible fashion and written in standard English?

Reviewer #1: Yes

Reviewer #2: Yes

Reviewer #3: Yes

Reviewer #1: The authors evaluated the effect of spinal needles in comparison to conventional needles for thyroid FNAB with respect to the rate of adequate diagnostic cytology as primary outcome in a multi-center, single-blind, randomized trial.

A logistic regression model was used to analyze the primary outcome with respect to the two types of needles. In accordance to the ICH E9 guideline “Statistical principles for clinical trials”, the main treatment effect should be investigated using a statistical model that allows for centre differences. I.e. the centre effect should be included in the statistical model of the primary analysis. Subsequently, the treatment by centre interaction should be evaluated in a sensitivity analysis and carefully discussed.

The significance level for all analyses was set to 5%. Considerations how multiplicity of analyses will be taken into account (e.g. appropriate adjustment of the significance level) should be provided.

Sample size calculation is one of the most important steps in the design of a clinical trial and should be presented in a transparent and understandable manner. The number of patients required has been calculated based on an expected improvement of 10% in the primary outcome, resulting in 175 patients in each group with a significance level of 5% and a power of 80%. This information is not sufficient for a clear understanding of the calculation. For example, if a chi-square test is used based on a 10% change from 75% to 85%, approximately 248 patients are needed, but for a change from 86% to 96%, 125 patients are needed. The number will be different for another statistical model, such as a stratified logistic regression model. Therefore, the used statistical model as well as the basic rate in addition to the expected change is a necessary piece of Information.

Furthermore, it must be pointed out that the assumptions for the sample size calculation differs betweeen the published protocol and the manuscript at hand.

It would also be helpful to explain why 380 patients were randomised although only 350 were to be included in the study.

In addition, contradictory data regarding the analyzed numbers of patients can be found in the flow-diagram (Figure 1, Analysed spinal needle 178, conventional needle 181) and in Table 2 (spinal needle 181, conventional needle 178).

Reviewer #2: The authors presented a very interesting article concerning the current topic regarding the choice of thyroid biopsy needle. Considering the large number of patients who undergo this procedure all over the world, the choice of needle - standard or spinal, that is, the result obtained by using one or the other, as well as the accompanying side effects and complications, is of great importance. Earlier studies showed the advantage of the spinal needle, however, this study shows that there is no such advantage, and when one takes into account the significantly higher price of the spinal needle compared to the standard needle, a clear conclusion is reached that the current practice of using the standard needle is satisfactory. The study was well designed and conducted and clearly presented. The conclusions are plain and applicable in everyday work. I am of the opinion that in the text of the article it is not necessary to state the exact names of the institutions and persons who performed any procedures, and that is the only correction that needs to be made.

Reviewer #3: While the randomization was centralized and pre-generated, it is not clear how allocation was concealed (e.g., via sealed opaque envelopes, secure web-based tool, third-party randomization center).

authors to explicitly describe the allocation concealment mechanism. This is critical to avoid selection bias.

Did not mention operator blinding or cytopathologist blinding, which should be clarified

Were pathologists assessing samples blinded to needle type?

Was any subgroup analysis performed for deep vs superficial nodules? This would greatly increase the clinical value.

Limitations section should be expanded:

1.Subjective pain measurement.

2.Operator variability (e.g., junior vs senior radiologist).

3.Blinding limitations.

Discussion should cite other RCTs (if any exist) or similar cohort studies and explain how this study adds new evidence.

**Do you want your identity to be public for this peer review?** For information about this choice, including consent withdrawal, please see our Privacy Policy

Reviewer #1: No

Reviewer #2: **Yes: ** Stanislav Rajkovic, M.D., PhD

Reviewer #3: No

---

## [Author Response · Author response to Decision Letter 1]

12 Jun 2025

Dear Editor,

We thank you and the reviewers for your time and the insightful comments on our manuscript, “Spinal Needles Versus Conventional Needles for Fine-Needle Aspiration Biopsy of Thyroid Nodules – A Multicenter Randomized Controlled Trial.”

We have carefully considered all the feedback and revised the manuscript accordingly. Below we provide a point-by-point response to each comment (with the responses quoted in red text for clarity). All changes to the manuscript are clearly highlighted in the revised version. We believe these revisions address the concerns raised and have improved the manuscript. We are grateful for the opportunity to clarify our work. Thank you again for your constructive input.

Reviewer #1 Comments:

Reviewer 1 – Comment 1: ”A logistic regression model was used to analyze the primary outcome with respect to the two types of needles. In accordance to the ICH E9 guideline “Statistical principles for clinical trials”, the main treatment effect should be investigated using a statistical model that allows for centre differences. I.e. the centre effect should be included in the statistical model of the primary analysis. Subsequently, the treatment by centre interaction should be evaluated in a sensitivity analysis and carefully discussed.”

Response 1: We appreciate this important comment. We agree that accounting for the multicenter design is essential. In fact, we did account for center effects in our analysis, but we recognize that our original description was not sufficiently explicit. To address this, we have modified the Statistical Analysis section to emphasize that we used a clustered modeling approach.

Specifically, we now write: “To adjust for the multicenter design, analyses of the primary outcome were conducted using a generalized estimating equation (GEE) logistic regression model with study center as a clustering variable. Similarly, clustering by operator was considered to account for any physician-related effects.” (Page 10, lines 213-217)

We also note in the results that this approach did not materially change the findings (the primary outcome results remained essentially the same after adjusting for center). By clarifying our use of a GEE model (with centers as clusters), we have addressed the reviewer’s concern. If needed, we are prepared to provide further details or additional analyses regarding center effects. We hope this explanation satisfies the reviewer.

Reviewer 1 – Comment 2: The significance level for all analyses was set to 5%. Considerations how multiplicity of analyses will be taken into account (e.g. appropriate adjustment of the significance level) should be provided.

Response 2: Thank you for highlighting the need to explain our strategy for type-I-error control. We prespecified one single primary endpoint, the proportion of adequate cytological samples and powered the trial exclusively for that comparison. Consequently, the family-wise error rate for efficacy is fully controlled at α = 0.05 for the primary analysis, and no further adjustment is required for that single test (ICH E9 §2.2.2).

All secondary (pain score, complications, diagnostic accuracy) and subgroup analyses were explicitly labelled exploratory in the original protocol and statistical-analysis plan. In line with CONSORT and ICH guidance, we therefore report these results with nominal two-sided p-values and 95 % confidence intervals for descriptive insight only, without formal multiplicity correction. To enhance transparency we have now added the following to the method and discussion:

“The overall trial alpha was fixed at 0.05 for the single, prespecified primary outcome. Secondary and subgroup analyses were exploratory; accordingly, their p-values are presented without multiplicity adjustment and any findings should be regarded as hypothesis-generating.” (Page 10, lines 233-236))

“Furthermore, because only one primary comparison was powered and multiplicity adjustment was not applied to exploratory analyses, positive signals among secondary outcomes should be interpreted with caution.” (Page 18, lines 400-402)

Reviewer 1 – Comment 3: Sample size calculation is one of the most important steps in the design of a clinical trial and should be presented in a transparent and understandable manner. The number of patients required has been calculated based on an expected improvement of 10% in the primary outcome, resulting in 175 patients in each group with a significance level of 5% and a power of 80%. This information is not sufficient for a clear understanding of the calculation. For example, if a chi-square test is used based on a 10% change from 75% to 85%, approximately 248 patients are needed, but for a change from 86% to 96%, 125 patients are needed. The number will be different for another statistical model, such as a stratified logistic regression model. Therefore, the used statistical model as well as the basic rate in addition to the expected change is a necessary piece of Information.

Response 3: We thank Reviewer #1 for pointing out the need for clarification on the sample size calculation. We have revised the Methods section to clearly explain our a priori power analysis and assumptions. In the revised manuscript, we now state:

“The sample size was determined based on an expected 10% absolute increase in adequate cytological diagnoses (from 86% in the control group to 96% in the spinal needle group, as suggested by prior studies (16,18). Using a two-sided α of 0.05 and power (1–β) of 90%, we calculated that 175 patients per group (350 total) would be required to detect this difference (19).” (Page 9, lines 197-201)

This added text details the effect size assumption, significance level, and power that underpinned our sample size calculation. We trust that this clarification addresses the reviewer’s concern.

Reviewer 1 – Comment 4: “It would also be helpful to explain why 380 patients were randomised although only 350 were to be included in the study.” In addition, contradictory data regarding the analyzed numbers of patients can be found in the flow-diagram (Figure 1, Analysed spinal needle 178, conventional needle 181) and in Table 2 (spinal needle 181, conventional needle 178).

Response 4:

We agree that clarification is warranted. Our target was to analyze 350 participants who actually underwent the allocated FNAB. To reach this figure we continued randomizing until 350 procedures had been completed, anticipating that a small proportion of randomized patients might not proceed to biopsy. In practice, 380 envelopes were opened, but 21 patients did not receive the intervention and therefore contributed no data. The most common reasons were:

• withdrawal of consent immediately after randomization (n = 9),

• clinical decision not to perform FNAB (n = 6)

• envelope opened in error for an ineligible patient (n = 6).

Thus, 359 participants received the intervention thereby exceeding the prespecified information size.

We therefore added following:

“The calculated sample size (n = 350) referred to participants who actually underwent the allocated FNAB. To compensate for anticipated post- randomization drop-outs (~10 %), we continued enrolment until ≥ 350 procedures had been completed. Enrollment stopped after 380 envelopes had been opened; 21 patients did not proceed to, leaving 359 analyzable participants.” (Page 9, lines 202-206)

We trust that this focused explanation resolves the reviewer’s concern.

Thank you for drawing our attention to this inconsistency. The numbers in Table 2 are correct (spinal needle = 181; conventional needle = 178). The values were inadvertently transposed in the final box of the CONSORT flow diagram.

• We have replaced Figure 1 with a corrected version that now reads

“Analyzed: spinal needle (n = 181); conventional needle (n = 178).”

Reviewer #2 Comments:

Reviewer 2 – Comment: The authors presented a very interesting article concerning the current topic regarding the choice of thyroid biopsy needle. Considering the large number of patients who undergo this procedure all over the world, the choice of needle - standard or spinal, that is, the result obtained by using one or the other, as well as the accompanying side effects and complications, is of great importance. Earlier studies showed the advantage of the spinal needle, however, this study shows that there is no such advantage, and when one takes into account the significantly higher price of the spinal needle compared to the standard needle, a clear conclusion is reached that the current practice of using the standard needle is satisfactory. The study was well designed and conducted and clearly presented. The conclusions are plain and applicable in everyday work. I am of the opinion that in the text of the article it is not necessary to state the exact names of the institutions and persons who performed any procedures, and that is the only correction that needs to be made.

Response: We thank the reviewer for the positive assessment of our work and for the helpful suggestion regarding textual economy.

Institutions: In the Methods section (“Study design and population”) we originally listed the full names of the three recruiting departments to document the multicentre setting. As this information may still be useful to readers evaluating external validity, we have kept the hospital names but streamlined the wording (e.g. “Department of Otorhinolaryngologyt” rather than the full official title and place).

Individuals: We have removed the names of individual physicians and technicians from the narrative text, retaining only those required in the author list and the brief acknowledgment of Ms Tina K. Agander for cytological slide preparation (as permitted by PLOS ONE). No procedural details now identify specific persons.

These edits shorten the manuscript and preserve confidentiality without compromising methodological transparency. We appreciate the reviewer’s thoughtful remark.

Reviewer #3 Comments:

Reviewer 3 – Comment 1: ”While the randomization was centralized and pre-generated, it is not clear how allocation was concealed (e.g., via sealed opaque envelopes, secure web-based tool, third-party randomization center). Authors to explicitly describe the allocation concealment mechanism. This is critical to avoid selection bias. Did not mention operator blinding or cytopathologist blinding, which should be clarified ”

Response 1: We appreciate this important observation. Allocation was rigorously concealed by using the sequentially-numbered, opaque, sealed-envelope, prepared and stored by personnel who had no role in patient enrollment. The two authors (TVA and KDL) created the center-stratified, block- randomization list (block size = 10) with www.randomization.com. A research secretary not involved in recruitment transferred each code to a tamper-evident, opaque envelope, numbered in chronological order. Each envelope contained: (i) the assignment (“spinal” or “conventional”), (ii) a colour-coded needle sticker so the operator could immediately select the correct device, and (iii) the participant ID label. The clinician opened the next envelope in sequence before the biopsy; thus neither investigators nor participants could foresee the assignment. Cytopathologists and data analysts remained blinded to group allocation throughout the study.

Randomization and Allocation Concealment:

We now explain that:

“Participants were randomly assigned (1:1) to the spinal needle group or the conventional needle group using a computer-generated random sequence with permuted block sizes (=10). The random sequence was generated by two investigators before the start of the trial using the website www.randomization.com. To ensure strict allocation concealment, the group assignments were sealed in sequentially numbered, opaque envelopes prepared before the start of the trial. Each envelope contained the designated needle (spinal or conventional) and was only opened by the clinician after the patient had been enrolled.” (Page 8, lines 174-182)

This clarification should reassure the reviewer that the randomization process was properly concealed and prevented any potential selection bias.

Blinding:

We have also clarified the blinding in our trial. The revised text reads:

“This trial was single-blinded. The patients were not informed about needle type used for their biopsy, and importantly, the cytopathologists were blinded as for the needle type. The clinicians performing the FNAB procedure, however, could not be blinded due to the obvious differences in needle appearance and handling, these operators were aware of the needle type upon opening the envelope. No other personnel had access to allocation information during the trial.” (Page 9, lines 188-194)

Reviewer 3 – Comment 2: “Was any subgroup analysis performed for deep vs superficial nodules? This would greatly increase the clinical value.”

Response 2: We appreciate the suggestion and agree that this could affect the diagnostic rate. However, no subgroup analysis by nodule depth was performed for the following reason: Depth was not systematically recorded for every participant at any of the three centers. Even if this were information available, the study was powered solely for the primary endpoint in the overall cohort, not for additional depth-stratified comparisons.

Reviewer 3 – Comment 3: “Limitations section should be expanded:

1. Subjective pain measurement.

2. Operator variability (e.g., junior vs senior radiologist).

3. Blinding limitations.”

Response 3: Thank you for this relevant comment. We have now added this to the limitations in the Discussion:

“This study has several limitations that warrant consideration. First, pain was assessed with a single, patient-reported 0–10 Numerical Rating Scale immediately after biopsy. Such subjective measurement is susceptible to individual pain-threshold variation. Second, although we adjusted for operator experience (> 4 years vs ≤ 4 years) in the multivariable models, other aspects of operator variability (e.g. specialty background, annual biopsy volume, junior vs senior status) could not be fully captured and may still influence cytological adequacy. Third, blinding was incomplete, patients and cytopathologists were blinded to needle type, but the performing clinicians could not be blinded because of obvious visual and tactile differences between the needles, introducing a potential performance bias. Fourth, A limitation of our study is that only 26.1% of patients referred for thyroid FNAB at the participating hospitals were enrolled in the study. This is mainly due to the SPINAB trial being conducted in a real-world clinical setting, where limited time and availability did not allow all patients to be invited. Further, only physicians who received instructions and hands-on training in the use of spinal needle FNAB could include patients. We prioritized the training of physicians in the spinal needle technique to ensure standardized performance in the spinal group. Consequently, we could not include all patients referred for FNAB. However, since our post hoc analysis did not find any differences between the included vs. non-included patients in the SPINAB trial, we believe the results are generalizable beyond the included patients (S4).” (Page 16-17, line 350-370)

Reviewer 3 – Comment 4: “Discussion should cite other RCTs (if any exist) or similar cohort studies and explain how this study adds new evidence.”

Response 4: To our knowledge, no previous randomized controlled trial has compared spinal versus conventional needles for thyroid FNAB. We have therefore reinforced the Discussion by contrasting our findings with the three largest prior non-randomized studies and explaining the incremental value of our RCT design:

“Prior to the current study, three non-randomized prospective studies by Cappelli et al. and Ahari et al. have been performed, including 516, 320, and 240 patients, respectively (16-18). These studies found that spinal needles reduce inadequate samples by 6–12 percentage points(16–18). However, those studies were single-center, relied on a single operator and lacked randomization, raising the possibility of selection and performance biases. Our multicenter RCT, involving 23 operators a

---

## [Decision Letter · Decision Letter 1]

Spinal Needles Versus Conventional Needles for Fine-Needle Aspiration Biopsy of Thyroid Nodules - A Multicenter Randomized Controlled Trial

PONE-D-25-08650R1

Dear Dr. Kasper Daugaard Larsen,

We’re pleased to inform you that your manuscript has been judged scientifically suitable for publication and will be formally accepted for publication once it meets all outstanding technical requirements.

Kind regards,

Shafiya Imtiaz Rafiqi, PhD

Academic Editor

PLOS ONE

Additional Editor Comments (optional):

Reviewers' comments:

Reviewer's Responses to Questions

**Comments to the Author**

Reviewer #1: All comments have been addressed

Reviewer #2: All comments have been addressed

2. Is the manuscript technically sound, and do the data support the conclusions?

Reviewer #1: (No Response)

Reviewer #2: Yes

3. Has the statistical analysis been performed appropriately and rigorously?

Reviewer #1: (No Response)

Reviewer #2: Yes

4. Have the authors made all data underlying the findings in their manuscript fully available?

Reviewer #1: (No Response)

Reviewer #2: Yes

5. Is the manuscript presented in an intelligible fashion and written in standard English?

Reviewer #1: (No Response)

Reviewer #2: Yes

Reviewer #1: (No Response)

Reviewer #2: The authors have addressed the comments. The paper is well written, has clear and clinically useful conclusion. It is good to publish in PLOS One.

**Do you want your identity to be public for this peer review?** For information about this choice, including consent withdrawal, please see our Privacy Policy

Reviewer #1: No

Reviewer #2: **Yes: ** Stanislav Rajkovic

---

## [Editor Report · Acceptance letter]

PONE-D-25-08650R1

PLOS ONE

Dear Dr. Larsen,

I'm pleased to inform you that your manuscript has been deemed suitable for publication in PLOS ONE. Congratulations! Your manuscript is now being handed over to our production team.

Kind regards,

on behalf of

Dr. Shafiya Imtiaz Rafiqi

Academic Editor

PLOS ONE